# The Neuro-Protective Effects of the TSPO Ligands CB86 and CB204 on 6-OHDA-Induced PC12 Cell Death as an In Vitro Model for Parkinson’s Disease

**DOI:** 10.3390/biology10111183

**Published:** 2021-11-15

**Authors:** Sheelu Monga, Nunzio Denora, Valentino Laquintana, Rami Yashaev, Abraham Weizman, Moshe Gavish

**Affiliations:** 1Ruth and Bruce Rappaport Faculty of Medicine, Technion—Israel Institute of Technology, Haifa 31096, Israel; sheelumonga@campus.technion.ac.il (S.M.); rami.yashaev@campus.technion.ac.il (R.Y.); 2Department of Pharmacy—Pharmaceutical Sciences, University of Bari “A. Moro”, Orabona, St. 4, 70125 Bari, Italy; nunzio.denora@uniba.it (N.D.); valentino.laquintana@uniba.it (V.L.); 3Felsenstein Medical Research Center, Sackler Faculty of Medicine, Tel Aviv University, Tel Aviv 6997801, Israel; aweizman@clalit.org.il; 4Research Unit, Geha Mental Health Center, Petah Tikva 4910002, Israel

**Keywords:** Parkinson’s disease, TSPO, CB86 and CB204, substantia nigra, cell death, dopaminergic neurons

## Abstract

**Simple Summary:**

**Aims and objectives:** For this study, we hypothesized that the two TSPO ligands CB86 and CB204 can inhibit cellular apoptosis and necrosis by in in vitro cellular PD model of undifferentiated PC12 cells exposed to 6-hydroxydopamine (6-OHDA, 80 µM). The two TSPO ligands CB86 and CB204 seem to suppress cell death of PC12 induced by 6-OHDA. The results may be relevant to the use of these two TSPO ligands as therapeutic options for neurodegenerative diseases like Parkinson disease (PD). **Results:** The two ligands normalized significantly (57% and 52%, respectively, from 44%; whereas the control was 68%) cell proliferation at different time points from 0–24 h. As compared to control, the red count was increased up to 57-fold whereas CB86 and CB204 inhibited to 2.7-fold and 3.2-fold, respectively. CB86 and CB204 inhibited also normalized the cell viability up to 1.8-fold after the exposure to 6-OHDA, as assessed by XTT assay. The two TSPO ligands also inhibited apoptosis significantly (1.3-fold for both) as assessed by apopxin green staining. **Conclusion:** It appears that CB86, CB204, and maybe other TSPO ligands are able to slow the progression of neurodegenerative diseases like PD.

**Abstract:**

Parkinson’s disease (PD) is a progressive neurodegenerative disorder which is characterized by the degeneration of dopaminergic neurons in substantia nigra (SN). Oxidative stress or reactive oxygen species (ROS) generation was suggested to play a role in this specific type of neurodegeneration. Therapeutic options which can target and counteract ROS generation may be of benefit. TSPO ligands are known to counteract with neuro-inflammation, ROS generation, apoptosis, and necrosis. In the current study, we investigated an in vitro cellular PD model by the assessment of 6-hydroxydopamine (6-OHDA, 80 µM)-induced PC12 neurotoxicity. Simultaneously to the exposure of the cells to 6-OHDA, we added the TSPO ligands CB86 and CB204 (25 µM each) and assessed the impact on several markers of cell death. The two ligands normalized significantly (57% and 52% respectively, from 44%; whereas the control was 68%) cell proliferation at different time points from 0–24 h. Additionally, we evaluated the effect of these two TSPO ligands on necrosis using propidium iodide (PI) staining and found that the ligands inhibited significantly the 6-OHDA-induced necrosis. As compared to control, the red count was increased up to 57-fold whereas CB86 and CB204 inhibited to 2.7-fold and 3.2-fold respectively. Necrosis was also analyzed by LDH assay which showed significant effect. Both assays demonstrated similar potent anti-necrotic effect of the two TSPO ligands. Reactive oxygen species (ROS) generation induced by 6-OHDA was also inhibited by the two TSPO ligand up to 1.3 and 1.5-fold respectively, as compared to 6-OHDA group. CB86 and CB204 inhibited also normalized the cell viability up to 1.8-fold after the exposure to 6-OHDA, as assessed by XTT assay. The two TSPO ligands also inhibited apoptosis significantly (1.3-fold for both) as assessed by apopxin green staining. In summary, it appears that the two TSPO ligands CB86 and CB204 can suppress cell death of PC12 induced by 6-OHDA. The results may be relevant to the use of these two TSPO ligands as therapeutic option neurodegenerative diseases like PD.

## 1. Introduction

Parkinson’s Disease (PD) is one of the most common progressive neurodegenerative disorders, movement disorder associated most often with advanced age and is related to dopaminergic neurons loss in substantia nigra (SN) [1,2]. The pathophysiological mechanisms of PD, as revealed in animal models and patients, including neurodegeneration along with neuro-inflammation, neuro-melanin degradation with aggregation of α-synuclein [3]. Based on a previous study, PC12 cells were used as an in vitro model of PD, by exposing them to the neurotoxic agent 6-OHDA [4]. In the present study, based on the neuroprotective effect of TSPO ligands [5], we investigated the such an effect in PC12 cellular model of PD. Several factors as oxidative stress, inflammation, toxicity of reactive oxygen species (ROS) products are thought to be involved in the cell death in SN [6,7]. Both neuroinflammation-related apoptotic and necrotic neuronal cell death were reported in various neurodegenerative disorders [6,8]. 

Exposure of dopaminergic neurons to the endotoxin 6-hydroxydopamine (6-OHDA) is associated with apoptotic cell death and is commonly used as in vivo/in vitro models of PD [9]. The toxic effect of 6-OHDA is achieved by its accumulation in catecholaminergic neurons, with subsequent oxidation and formation of semiquinone radicals which have toxic effect on catecholaminergic neurons [10,11]. In our present study, we used the catecholaminergic PC12 cells as an in vitro model of dopaminergic cells relevant to PD. These cells were chosen since they do express TSPO [12].

The 18 kDa translocator protein (TSPO) is located mainly on the outer mitochondrial membrane and plays indispensable role in several cellular processes, including: mitochondrial function, steroid hormone regulation, inflammation response, and apoptotic pathways [13,14,15]. TSPO is abundant in many peripheral organs especially in steroidogenesis tissues as well as in glial cells in central nervous system (CNS) [16]. TSPO regulates the release of cytochrome c, which is associated with mitochondrial apoptotic cell death [17]. Moreover, TSPO is involved in cellular ROS production, which take part in apoptosis [18]. Since PD pathophysiology is related to apoptosis and neurotoxicity (necrosis) [19], it is reasonable to assume that TSPO might be involved in the pathogenesis of PD, as was shown by TSPO overexpression while dopaminergic neurodegeneration occurs [20]. Mitochondrial destruction and consequent apoptosis is implicated with other neurodegenerative disorders such as Alzheimer disease and Huntington’s disease [13]. It is of note that a previous in vivo study demonstrated the neuroprotective effect of TSPO ligands in 1-methyl-4-phenyl-1,2,3,6-tetrahydropyridine (MPTP) mouse model of PD [3].

We have shown previously the anti-inflammatory effects of the TSPO ligands CB86 and CB204 in glial cells [21]. Therefore, it was hypothesized in the present study that these TSPO ligands may be a promising novel target in the treatment of PC12 cellular model of PD. To this end, we assessed the in vitro neuroprotective effect of the TSPO ligands CB86 and CB204 (synthetic chemical compounds, the Ki value is higher towards the TSPO) in PC12 cells exposed to 6-OHDA as a cellular model for PD. 

## 2. Methods

### 2.1. PC12 Cells

Undifferentiated PC12 cells (a kind gift from Prof. Kobi Rosenblum, University of Haifa, Haifa, Israel), derived from rat adrenal pheochromocytoma, were used as in vitro cellular model for PD in this study. 6-hydroxydopamine, a structural analog of dopamine, has neurotoxic effect on PC12 cells and thereby represents a model for apoptosis and necrosis of dopaminergic neurons, which might be relevant to pathophysiology of PD [22].

The PC12 cells were cultured at 37 °C with 5% CO_2_ and 90% relative humidity. Cells were grown in Dulbecco’s modified Eagle’s medium (DMEM) with 2 mM L-glutamine, 40% fetal calf serum, 40% donor horse serum and 1% of pen-strep (penicillin-streptomycin) with nystatin. 

### 2.2. 6-Hydroxidopamine Exposure and TSPO Ligands Treatment

2.5 × 10^4^ undifferentiated PC12 cells were seeded in 24-well plate and then incubated for 48 h until the confluency of 50–60% was reached (using light inverted microscope, Leica, Buffalo Grove, IL, United States) and then subsequently exposed to 6-OHDA (80 µM) as well as treated simultaneously with the TSPO ligands CB86 and CB204 (25 µM each) for next 24 h. This concentration of 6-OHDA was chosen based upon a dose-response study (Appendix A). The concentration of the TSPO ligands CB86 and CB204 (25 µM each) was used in our previous study [23].

### 2.3. Trypan Blue Staining for Cell Counting

The cells were trypsinized [Trypsin EDTA Solution B (0.25%), EDTA (0.05%)] (Biological industries, Beit Ha’Emek, Israel) before seeding into 24-well plate. The procedure was performed by the mixture of cells and trypan blue dye (1:1) in an Eppendorf tube. Ten microliters of this mixture were used on the specific Neubauer’s slide and covered by cover slip. The cells were counted manually by the hemocytometer and the number of the cells was recorded and calculated per milliliter. 

### 2.4. Necrotic Analysis with Fluorescence Microscopy

We performed microscopic fluorescent analysis using Incucyte Zoom microscope (Essen Bioscience, Ann Arbor, Michigan). As a fluorophore, propidium iodide (PI) (1 mg/mL) was used and this stock was diluted to 1:1000 to stain the necrotic cells using wavelength Ex/Em = 482/608 nm and the necrotic cells emit red fluorescence color upon staining. 

### 2.5. Cell Viability Assay

We performed cell viability assay, using Cell Proliferation Kit II (XTT) (Biological Industries, Beit Ha’Emek, Israel). The 2,3-bis[1-methoxy-4-nitro-5-sulphonyl]-2H-tetrazolium-5-carboxyanilide inner salt (XTT) is based on the reduction of XTT by mitochondrial dehydrogenases of viable cells yielding an orange formazan product. Optic density (O.D.) was measured using Infinite M200 Pro plate reader (Tecan, Männedorf, Switzerland) with absorbance at endpoint photometric of 492 nm main wavelength and 620 nm reference wavelength.

### 2.6. LDH Cytotoxicity Assay

Cytotoxicity assay using an LDH Cytotoxicity Detection kit (Merck KGaA, Darmstadt, Germany) was performed. After 24 h of treatment, the cell culture supernatant (100 µL) was collected and transferred into another 96-well plate. LDH reagent, prepared according to the manufacturer’s instructions, was added to each well (100 µL). After adding LDH reagent the plate was incubated on a plate shaker for 10 min and optical density (O.D.) was measured using Infinite M200 Pro plate reader (Tecan, Männedorf, Switzerland) with absorbance at endpoint photometric of 492 nm main wavelength and 620 nm reference wavelength.

### 2.7. ROS Production Assay

Intracellular ROS were analyzed by using ROS/Superoxide Detection Assay Kit (Abcam, Cambridge, UK). Briefly, after 24 h of 6-OHDA exposure with/without TSPO ligands (25 µM each), the supernatant was removed, and cells were washed by 1× Wash buffer and 100 µL of ROS/Superoxide detection solution was added, and cells were incubated for 1 h in the dark at 37 °C. Finally, the fluorescence intensity was recorded by fluorescent Infinite plate reader M200 Pro (Tecan, Männedorf, Switzerland) at wavelength Ex/Em = 482/608 nm.

### 2.8. Apoptosis Analyses

Apopxin green dye from necrosis/apoptosis assay kit (Abcam, Cambridge, UK) was used to detect apoptosis levels in PC12 cells according to the manufacturer’s instructions. Seeding of PC12 cells was performed in 12-wells plates (50,000 cells/well) and grown normally for 48 h in complete serum containing medium. After 24 h of exposure to 6-OHDA (80 µM) with/without CB86 and CB204 (25 µM each), cells were washed with 200 µL assay buffer (provided in the kit). Staining was conducted by adding 2 µL of Apopxin to 100 µL of sample for apoptotic cells detection, with subsequent incubation in dark for 1 h. For analyses, fluorescent plate reader Infinite M200 Pro (Tecan, Männedorf, Switzerland) at wavelength Ex/Em = 490/525 nm was used for detection of apoptotic levels with median fluorescence intensity.

### 2.9. Statistical Analyses

Results are presented as Mean ± standard deviation (SD) or standard error mean (SEM). One-way analysis of variance (ANOVA) test was used as appropriate, including Bonferroni’s post-hoc test. Statistical significance was defined by *p* < 0.05.

## 3. Results

### 3.1. Phase Confluence

PC12 cells were seeded in 24-well plate and exposed to 6-OHDA (80 µM) with or without CB86 or CB204 (25 µM each) for 24 h and stained with PI. Phase confluence and median fluorescence intensity was analyzed using Incucyte Zoom fluorescence microscope (Essen Bioscience, Ann Arbor, Michigan) every 2 h intervals from 0–24 h. The confluency at 0 h was started with 50–60% for this experiment. As shown in Figure 1, the phase confluence is highest in control after 24 h was ~70% but reduced significantly in 6-OHDA group ~40%. In 6-OHDA+CB86 and 6-OHDA+CB204 group, the confluency was better than 6-OHDA group, but not as good as control. In this experiment, CB86 showed slightly better effect than C B204 with respect to phase confluence. The two TSPO ligands CB86 and CB204 did not show any effect on their own.

### 3.2. Cell Necrosis with PI

Cell necrosis assay was performed to analyze the protective effect of the TSPO ligands CB86 and CB204 from 6-OHDA-indued cell necrosis. As shown in Figure 2A,B, the red object count was increased by 6-OHDA, and this elevation was counteracted significantly (*p* < 0.001) by both CB86 and CB204 (at a concentration of 25 µM each). The protective effect of the two ligands was similar. In Figure 2B, it shows that the red object count (PI contrast/staining, a marker of necrosis) reached its maximal level after 6 h of exposure to 6-OHDA and remained stable for the next 18 h. The two TSPO ligands exhibited a stable significant protective effect, a decrease in cellular necrosis, for the same 18 h. In this experiment, CB86 have shown slightly better effect than CB204 but this difference was not significant. The two ligands did not show any effect on their own.

### 3.3. Cell Cytotoxicity Assay with LDH

The cell cytotoxicity was induced by exposing PC12 cells to 80 µM 6-OHDA and treated simultaneously with or without CB86 or CB204 (25 µM each) for 24 h. The cell cytotoxicity was measured using LDH kit (Merck KGaA, Darmstadt, Germany). Exposure to 6-OHDA by itself induced significant elevation of LDH (25.9% compared to control), and this elevation was counteracted by the two TSPO ligands. CB86 showed significantly (*p* < 0.001) a better protective effect (15% compared to 6-OHDA) than CB204 (8% compared to 6-OHDA) (Figure 3). The two TSPO ligands did not have any effect on their own.

### 3.4. Cellular Viability Assay

Figure 4 shows that the cell viability/survival, as accessed by XTT kit (Biological Industries, Beit Ha’Emek, Israel), was significantly reduced with the exposure of PC12 cells to 6-OHDA (61.7% compared to control). The cells survival/viability was increased by the two TSPO ligands (29.8% and 28.4% with CB86 and CB204, respectively, compared to 6-OHDA; *p* < 0.001 for both. These TSPO ligands did not show any effect on their own as compared to control.

### 3.5. ROS/Superoxide Assay

As shown in Figure 5, in 6-OHDA group, the fluorescence intensity was significantly higher than control group (57.5%, *p* < 0.001) and it was counteracted by CB86 and CB204 (25 µM each) 23.5% and 36%, respectively, compared to 6-OHDA and *p* < 0.001 for both). In this experiment, CB204 showed slightly better effects than CB86 which was significantly different than 6-OHDA+CB86. None of the TSPO ligands showed effects on their own.

### 3.6. Apoptosis Assay

Apoptotic analysis was performed using apopxin green dye. After the PC12 cells were exposed to 80 µM 6-OHDA and simultaneously treated with or without the two TSPO ligands CB86 and CB204 (25 µM each). After staining with apopxin green dye, the analysis was performed using fluorescence microplate reader Tecan M200 pro (Tecan, Männedorf, Switzerland). Figure 6 shows that the level of fluorescence intensity was elevated in 6-OHDA group (33% compared to control) which was counteracted by CB86 and CB204 (16% compared to 6-OHDA) respectively. Both TSPO ligands have shown almost the same effect. The two TSPO ligands did not show any effect on their own.

## 4. Discussion

The present study investigated a novel approach for treating PD from the perspective of possible TSPO involvement in the pathophysiology of the disease. Therefore, this study was designed to explore the in vitro neuroprotective effects of the novel TSPO ligands CB86 and CB204 on 6-OHDA-induced PC12 cell death with respect to apoptosis, necrosis, and ROS generation.

6-OHDA oxidation by monoamine oxidase A (MAO-A) leads to the production of hydrogen peroxide which results in ROS generation, apoptosis/necrosis [10,24]. The PC12 cells which are derived from rat adrenal pheochromocytoma, represents monoaminergic neurons that are relevant to dopaminergic neurons [22,25]. The main mechanism of toxicity caused by 6-OHDA is the induction of ROS by autoxidation [9,26]. Thus, the PC12 cells may serve as a suitable model for exploring the putative neuroprotective effects of the TSPO ligands CB86 and CB204.

TSPO plays important role in apoptotic pathways [27]. It has been shown that TSPO expression was upregulated upon the activation of apoptotic pathways, may be as a protective cellular defense mechanism [28]. In our study, the two TSPO ligands CB86 and CB204 showed neuroprotective effects in several assays. As 6-OHDA is already known to suppress the cellular viability [28], phase confluence percent in the 6-OHDA treated cells, was dramatically reduced, but the addition of the two TSPO ligands inhibited this reduction (Figure 1). Noteworthy, these two TSPO ligands have shown previously protective effects in cells exposed to endotoxins [23]. Also, these TSPO ligands attenuated 6-OHDA-induced necrosis, as assessed by LDH release and red object count, in comparison to the control group. These two TSPO ligands have shown their complete ability of inhibiting necrosis within 6 h (Figure 2A). Additionally, the cellular viability (XTT) assay has shown significant neuroprotective effects of the two TSPO ligands in the cells exposed to 6-OHDA [10] (Figure 4). Apoptotic analysis also indicated high apoptotic levels of PC12 cells exposed to 6-OHDA, and TSPO ligands inhibited this 6-OHDA-related apoptosis [29]. Oxidative stress is the main cause of early apoptosis or necrosis in the body [30] and intracellular ROS generation by 6-OHDA is an initial event, in which ROS suppresses the Akt phosphorylation, increases p38 phosphorylation which induces the activation of caspases, and finally leads to cell apoptosis [31]. The ROS production assay have pointed a putative antioxidant activity of the TSPO ligands CB86 and CB204. As it is mentioned before, 6-OHDA is thought to cause ROS-generation [31], and TSPO ligands showed the reduction of such ROS production. Despite the promising data that clearly shows the neuroprotective effect of these two TSPO ligands in PC12 cells exposed to 6-OHDA, the relevance to human PD is unclear. Furthermore, it is possible that the use of dopaminergic cultured cells is a more appropriate cellular model. Nevertheless, in vivo studies are needed to confirm the beneficial effect of the TSPO ligands in animal models of PD.

## 5. Conclusions

The results indicated that the two TSPO ligands can inhibit not only 6-OHDA-induced ROS generation, but also apoptosis and necrosis of PC12 cells as well. Besides, these two TSPO ligands normalized the cellular viability and showed significant positive effects on cell viability and proliferation. Thus, it appears that CB86, CB204, and maybe other TSPO ligands are able to slow the progression of neurodegenerative diseases like PD. Such options merits further in vivo studies in appropriate animal models.

## Figures and Tables

**Figure 1 biology-10-01183-f001:**
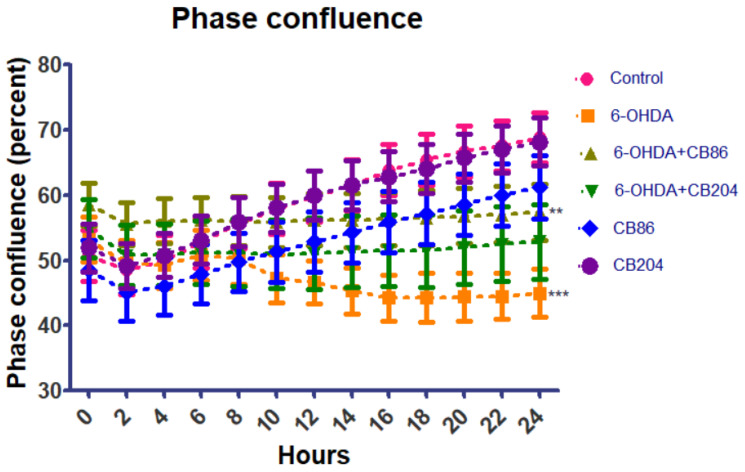
Phase confluence analysis: Phase confluence analysis was performed using Incucyte Zoom microscope in the different groups at 2 h intervals for 24 h. Results are expressed by mean ± SEM (*n* = 4 replicates in each group). ANOVA followed by Bonferroni’s post-hoc test was performed, *** compared to control and ** compared to 6-OHDA group, *p* < 0.001 for both.

**Figure 2 biology-10-01183-f002:**
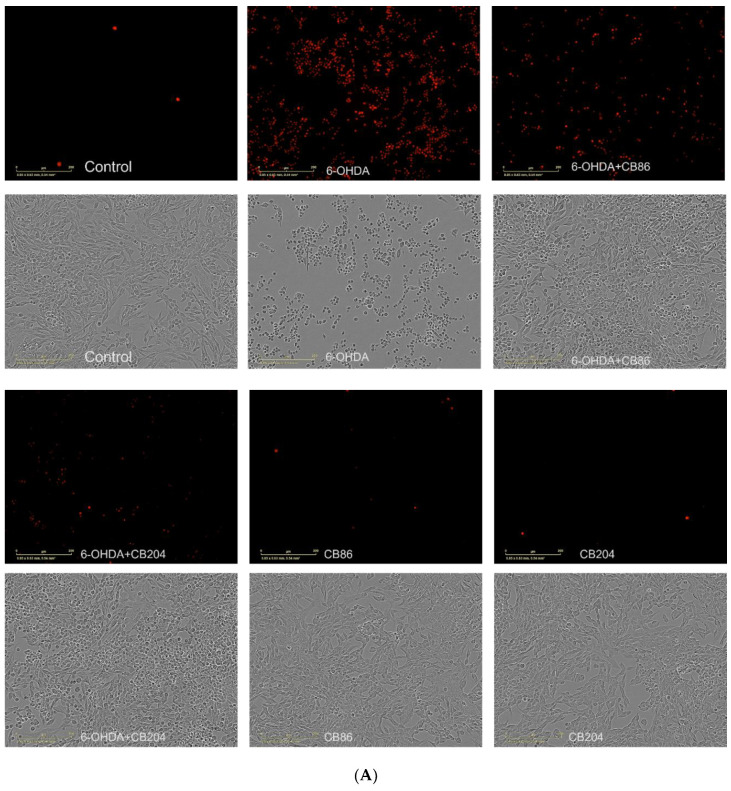
Cell necrosis assay with PI: PC12 cell necrosis was induced by exposure to 80 µM 6-OHDA and simultaneously treated or untreated with CB86 or CB204 (25 µM each) for 24 h. (**A**) shows microscopic images phase confluence vs PI with scale bar 0–200 µm and (**B**) red object count from 0–24 h. Results are expressed by mean ± SEM (*n* = 4 replicates in each group). ANOVA followed by Bonferroni’s post-hoc test was performed, ^###^ compared to control; *** compared to 6-OHDA group alone, *p* < 0.001 for both.

**Figure 3 biology-10-01183-f003:**
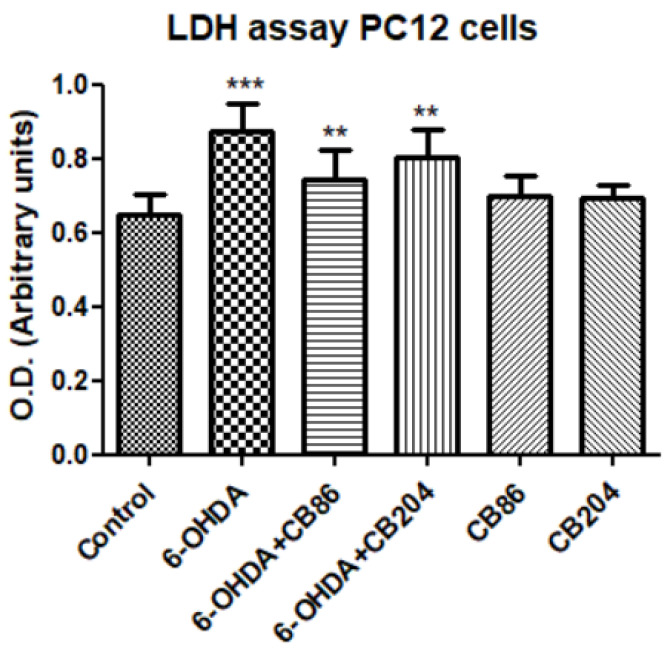
Cell cytotoxicity assay with LDH kit: cell cytotoxicity was induced by exposure of PC12 cells to 80 µM 6-OHDA for 24 h. Simultaneously, the PC12 cells were treated with CB86 or CB204 (25 µM each) for 24 h. Results are expressed by mean ± SD (*n* = 16 in control and *n* = 8 replicates in all other groups). ANOVA followed by Bonferroni’s post-hoc test was performed, *** compared to control, ** compared to 6-OHDA and 6-OHDA+CB204; *p* < 0.001 for all.

**Figure 4 biology-10-01183-f004:**
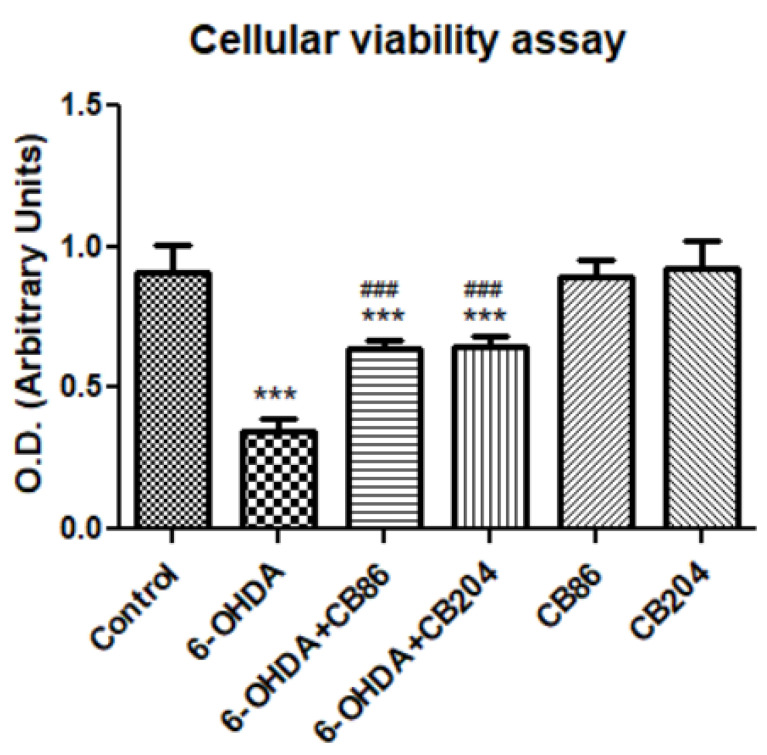
Cell viability assay with XTT: cell viability of PC12 was affected by the exposure of 80 µM 6-OHDA for 24 h. When the cells exposed simultaneously to 6-OHDA with TSPO ligands CB86 or CB204 (25 µM each) for 24 h, the rate of cell survival was increased. Results are expressed by mean ± SD (*n* = 8 replicates in each group). ANOVA followed by Bonferroni’s post-hoc test was performed, F (5, 42) = 91.85, *p* < 0.001, *** vs. control and ^###^ compared to 6-OHDA, *p* < 0.001 for all comparisons.

**Figure 5 biology-10-01183-f005:**
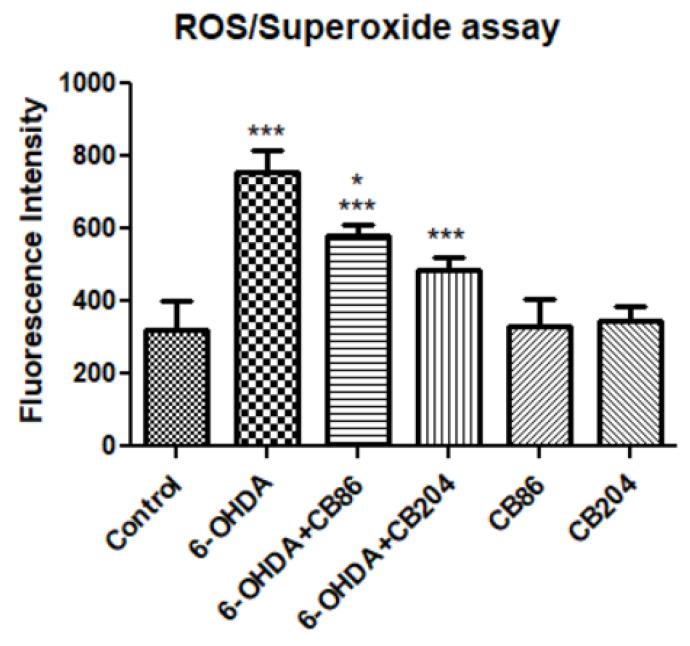
ROS/Superoxide assay: Release of ROS/superoxide was induced by the exposure of PC12 cells to 80 µM 6-OHDA for 24 h. Simultaneously, the PC12 cells were treated with or with CB86 or CB204 (25 µM each) for 24 h. Results are expressed by mean ± SD (*n* = 8 replicates in each group). ANOVA followed by Bonferroni’s post-hoc test was performed, F (5, 42) = 75.18, *p* < 0.001. *** *p* < 0.001 vs. control, and * *p* < 0.05 compared to 6-OHDA+CB204.

**Figure 6 biology-10-01183-f006:**
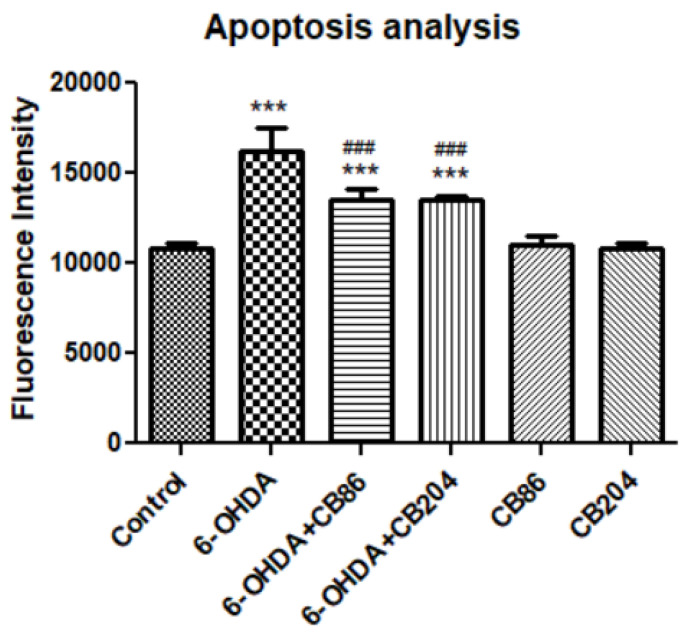
Apoptosis assay: Apoptosis was induced by the exposure of PC12 cells to 80 µM 6-OHDA for 24 h. The PC12 cells were treated simultaneously with CB86 or CB204 (25 µM each) for 24 h. Results are expressed by mean ± SD (*n* = 8 replicates in each group). ANOVA followed by Bonferroni’s post-hoc test was performed, F (5, 42) = 85.07, *p* < 0.001. *** compared to control and ^###^ compared to 6-OHDA, *p* < 0.001 for all comparisons.

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
