# Peer review of "The Neuro-Protective Effects of the TSPO Ligands CB86 and CB204 on 6-OHDA-Induced PC12 Cell Death as an In Vitro Model for Parkinson’s Disease"

_biology, 2021, doi:10.3390/biology10111183_

Round 1

Reviewer 1 Report

Monga et al reported the protective effects of two TSPO ligands in a PC12 cell injury model induced by 6-OHDA, providing valuable insight to develop therapeutic reagents for parkinson's disease.  Some concerns and questions are shown below.
  1. An important concern is the novelty of this work. The authors ignored a previous in vivo study showing neuroprotective functions of TSPO ligands in PD (reference 3; DOI: https://doi.org/10.1523/JNEUROSCI.2070-18.2019), although the ligands and model are different. More experiments targeting pro-death/pro-survival signaling might be considered.
  2. In the introduction, brief introduction of CB86 and CB204 is necessary.
  3. Which type of PC12 cells used? high differentiation, low differentiation or undifferentiating? Different types have distinct morphological pattern and physiological characteristics.
  4. What is the rational for the concentration selection of 6-OHDA (80 µM), CB86 and CB204 (at a concentration of 25 µM each)?
  5. What is the density of PC12 cells seeded in 24-well plate? Are the cells treated with drugs immediately after seeding? If so, the confluence at 0 h is not convincing. How to measure the confluence? The details are needed.
  6. Fig1: Dash line and hollow shape are recommended to make the figure clearer. Phase contrast images of PC12 cells in different conditions have to be provided for indicating morphological changes.
  7. Fig2: scale bar labelling too small to see. DAPI/PI double staining or Phase contrast/PI staining is necessary and important to reflect the total cell number and the survival rate in each condition.
  8. The limitations of using PC12 as a model for parkinson's disease research need to illustrate.

Author Response

Dear Reviewer,

Thank you for your kind and valuable comments on our manuscript. We have made few changes as per suggested your suggestions.

  1. The mentioned paper has been cited.
  2. Introduction was modified.
  3. Information about PC12 cells was added.
  4. For the concentration selection, we have added the supplementary file along with this manuscript.
  5. Number of cells and confluency was added.
  6. Phase contrast images have been added in the manuscript.
  7. Phase contrast images were added.
  8. Elaborated information about PC12 cells was added to the introduction.

Reviewer 2 Report

In this study, the authors used 6-OHDA to treat cultured PC12 cells to induce an in vitro PD model and observed the effects of two TSPO ligands, CB86 and CB204, on cell growth. The authors found that these two TSPO ligands can inhibit 6-OHDA-induced PC12 cell death. I have a few questions as follows.

Questions:

  1. Typically, as the authors mention in the introduction, cultured dopaminergic neurons exposed to 6-OHDA are used as in vitro models of PD. However, the authors used cultured PC12 cells exposed to 6-OHDA to prepare PD models. Can the authors explain why they did not use cultured dopaminergic neurons for experiments?
  2. How did the author determine the concentrations of 6-OHDA and two TSPO ligands used to treat the cells? Will different concentrations of drugs lead to different growth states of cells?
  3. The authors should provide representative images of cultured cells for Figure1, 3-6 to make the data more convincing.
  4. Is the 18 kDa translocator protein expressed in dopaminergic neurons?

Author Response

Dear Reviewer,

Thank you for your kind and valuable comments for our manuscript. We have modified the manuscript as per your suggestion.

  1. Explanation for using PC12 cells have been added.
  2. For 6-OHDA, we have uploaded the dose-response supplementary data with the manuscript and for ligands, it was calibrated in our previous study. That paper has been cited.
  3. Images have been modified for Fig. 1, and microscopy analysis was not performed in case of Fig. 3-6 but we used the fluorescent plate reader instead. The median fluorescence intensity/absorbance was recorded.
  4. Yes.

Round 2

Reviewer 1 Report

The concerns raised by the reviewer were not fully addressed. The authors should answer the questions one by one seriously and leave the response under each comment. Some unrelated answers should be evitable. 

1.The mentioned paper has been cited.

Please check comment 1. If the authors suppose no more experiments needed, please provide some rationales or solutions for review’s concern rather than just ignore the comments or give such a disappointing response.

2.Introduction was modified.

Please state the line number for the revised content for the comments.

3.Information about PC12 cells was added.

Please state the line number for the added content for the comments.

4.For the concentration selection, we have added the supplementary file along with this manuscript.

No related information was found in supplementary file.

5.Number of cells and confluency was added.

How to measure the confluence? The details are needed.

The author stated the cell density and confluence (80%) (line 90) but this statement is against the confluence at 0 h in fig1.

6.Phase contrast images have been added in the manuscript.

Here is the comment 6 : “Fig1: Dash line and hollow shape are recommended to make the figure clearer. Phase contrast images of PC12 cells in different conditions have to be provided for indicating morphological changes.” 

Phase contrast images are easy to aquire and are the basic evidence for morphological changes. There is still no sample images were shown in fig1.

7.Phase contrast images were added.

Here is the comment 7: “ Fig2: scale bar labelling too small to see. DAPI/PI double staining or Phase contrast/PI staining is necessary and important to reflect the total cell number and the survival rate in each condition.”

In fig2, the author showed Phase contrast/PI staining. It appears most cells in 6-OHDA group were dead. This result is inconsistent with fig4. The survival rate in each condition has to be provided.

8.Elaborated information about PC12 cells was added to the introduction.

The limitations of using PC12 as a model for parkinson's disease research need to illustrate in discussion.

Author Response

The response to the reviewer's comments are attached as a separate file.

Reviewer 2 Report

The authors answered some of the questions with satisfactory responses, but there are still some questions that need further explanation.

  1. The authors did not explain why 6-OHDA was not used to induce cultured dopaminergic neurons for PD model. In addition, the authors mentioned in their revised manuscript that "In our present study, we used the catecholaminergic PC12 cells as an in vitro model of dopaminergic cells relevant to PD [10] ", However, this reference does not mention that 6-OHDA can induce PD model at all.
  2. The concentrations of TSPO Ligands mentioned by the author were determined based on previous studies. However, the ref. 24 shows that BV-2 microglial cells were used in the study instead of PC12 cells. Therefore, the authors did not explain this issue clearly.

Author Response

The response to the reviewer's comments is attached as a separate file.

Round 3

Reviewer 1 Report

What did the scale bar indicate in fig2? 100um?

The inconsistence is in fig2 6-OHDA group almost all the cells dead,so the cellular viability should decrease to 10% or less. But in fig4 cellular viability was decreased about 60% compared with control. How to explain this? Survival rate of the cells (PI negative cell number/ total cell number) in each group are strongly recommended.

Why did the author change the confluency percentage without any mentions in the response?

In last version Line89: 2.5*104 undifferentiated PC12 cells were seeded in 24-well plate and then incubated for 48 hours until the confluency of 80% was reached (using light inverted microscope) In this version line92: 2.5*104 undifferentiated PC12 cells were seeded in 24-well plate and then incubated for 48 hours until the confluency of 70% was reached (using light inverted microscope, Leica, Buffalo Grove, IL, United States)

Line92: 2.5*104 undifferentiated PC12 cells were seeded in 24-well plate and then incubated for 48 hours until the confluency of 70% was reached (using light inverted microscope, Leica, Buffalo Grove, IL, United States) and then subsequently were exposed to 6-OHDA (80 µM) as well as treated simultaneously with the TSPO ligands CB86 and CB204 (25 µM each) for next 24 hours. This statement indicated they started drug exposure experiments when the cell confluency reached 70%. So the 70% should be the value for t=0 in fig1. But they said the confluency at 0 hour was started with 50-60% for this experiment in line 162. Which one is correct?

The representive image for Fig1 is needed (at least t=0 and t=24, otherwise the data is not convincing). The authors said they provided phase contrast images in Fig 2A which also can indicate the confluency changes in fig1 during 24 hours. If that is the case, 6-OHDA group confluency should be 45% in fig2 but from the sample image this value should be much less than 45%.

Author Response

The author's response is mentioned in the file attached.

Reviewer 2 Report

The authors have addressed all my concerns. I have no further questions. Thank you. 

Author Response

Thank you for your kind response approving the manuscript.